# Community Perceptions of Alcohol Exposed Pregnancy Prevention Program for American Indian and Alaska Native Teens

**DOI:** 10.3390/ijerph16101795

**Published:** 2019-05-21

**Authors:** Umit Shrestha, Jessica Hanson, Tess Weber, Karen Ingersoll

**Affiliations:** 1Department of Community and Behavioral Health, Centers for American Indian and Alaska Native Health, Colorado School of Public Health, University of Colorado Anschutz Medical Campus, 13055 East 17th Avenue, Aurora, CO 80045, USA; 2Behavioral Sciences, Sanford Research, 2301 East 60th Street North, Sioux Falls, SD 57104, USA; jessica.d.hanson060979@gmail.com (J.H.); tess.weber@sanfordhealth.org (T.W.); 3University of Virginia School of Medicine, 310 Old Ivy Way, Charlottesville, VA 22903, USA; kes7a@virginia.edu

**Keywords:** alcohol-exposed pregnancy, American Indian, teens/adolescents, community perception, qualitative research

## Abstract

A community needs assessment during a tribally-led Changing High-Risk Alcohol Use and Increasing Contraception Effectiveness Study (CHOICES) intervention highlighted the need to reduce the risk for alcohol exposed pregnancy (AEP) among American Indian and Alaska Native (AIAN) adolescent girls. The CHOICES for American Indian Teens (CHAT) Program aims to reduce the risk of AEP among AIAN teens in one Northern Plains tribal community. The CHAT team adopted an iterative process to modify the tribally-led CHOICES curriculum for AIAN teens. This paper describes the iterative process as well as the community perception towards AEP prevention among AIAN teens. The CHAT team conducted several levels of formative and qualitative research, including one-on-one interviews (n = 15) with community members, AIAN elders and school counsellors; and three focus groups with AIAN adolescent girls (n = 15). A qualitative data analysis identified several recommendations that centered on making the information regarding alcohol and birth control appealing to teens; ensuring the confidentiality of the participants; making the program culturally relevant; and including boys in the program. This study outlines various components prioritized by community members in creating a culturally-relevant and age-appropriate AEP prevention program and provides community perceptions of AEP prevention for the teens in this community

## 1. Introduction

Fetal Alcohol Spectrum Disorder (FASD) is the umbrella term that denotes the continuum of disabilities resulting from prenatal alcohol exposure. Fetal Alcohol Syndrome (FAS) is the most recognized and most severe impact of prenatal alcohol consumption. FAS has damaging effects, including poor growth, small head size, and facial abnormalities, along with the potential for the delayed development of important organs in the fetus and of social and behavioral issues later in life [1,2,3]. Other diagnoses on the FASD spectrum include partial FAS, alcohol-related neurodevelopmental disorder, and alcohol-related birth defects [1,2]. FASD remains one of the leading preventable neurodevelopmental disorders in the world [4].

It is difficult to obtain community-specific FASD prevalence data for a variety of reasons, including issues with the self-reporting of maternal drinking during pregnancy, sampling, and a lack of consistent diagnostic criteria and routine screenings in prenatal clinics [5,6,7]. One recent study demonstrated that the prevalence estimates for FASD among younger schoolchildren in the United States range from 11.3 to 50.0 per 1000 children [8]. The Northern Plains American Indian and Alaska Native (AIAN) has one of the highest rates (9 per 1000 live births) of FAS in the world [9]. Similarly, prenatal alcohol consumption is also higher among select samples of AIAN women [10,11,12]. Such high FAS rates and risky drinking patterns couple to make FASD a major public health concern among the AIAN population. 

The pre-conceptional prevention of alcohol-exposed pregnancies (AEP) focuses on preventing unintended pregnancies among women who are drinking at binging levels or on reducing drinking in women at risk of pregnancy, thereby applying a dual-behavior approach to reduce the risk for AEP [13,14]. Changing High-Risk Alcohol Use and Increasing Contraception Effectiveness Study (CHOICES) is an evidenced-based program that focuses on the pre-conceptional prevention of AEP by employing multiple in-person sessions over time [15,16,17,18,19]. CHOICES utilizes motivational interviewing to guide participants to resolve their ambivalence by increasing the perceived discrepancy between behaviors and values [18]. It uses various activities, such as decisional balance exercises, goal setting and personalized feedback, to discuss change plans regarding the reduction of drinking and the prevention of an unplanned pregnancy, and it encourages participants to take ownership of the changes they wish to induce within their lifestyle [17,18]. The CHOICES intervention has been successful in different settings and has been replicated in various communities, including AIAN communities, with success in reducing the risk for AEP among adult women [20,21,22].

Work has also been done to understand how AEP prevention can work with AIAN teens. In general, AEP prevention with teens is important for the same reasons as it is for adult women. One of the few studies on teen substance use during pregnancy found that pregnant teens report elevated levels of substance use prior to becoming pregnant, with varying substance uses by subgroups of pregnant adolescents [23,24]. A national study from 2012 found that pregnant teens reported a greater substance use (18.3%) compared with same-age non-pregnant peers (13.8%), and twice the rate of pregnant youths aged 18 to 25 years (9%) [25]. A study from 2015 found that 24.9% of pregnant teenagers reported using one or more substances in the past 30 days, although the majority appeared to have quit using substances when they found out that they were pregnant [24].

The risk for AEP among AIAN youths is likely high because of several risk factors. National data found that 22.9% of AIAN youths aged 12 and older reported alcohol use, 18.4% reported binge drinking, and 16.0% reported substance dependence or abuse [26]. American Indian (AI) adolescents are also more likely to have consumed alcohol in their lifetime compared to Caucasian adolescents and have higher rates of binge drinking compared to Caucasian adolescents [27,28,29,30]. The rate of lifetime alcohol use among AI girls in grades 7–12 was nearly 71%, significantly higher than the rates reported by Caucasian girls (56%) and AI boys (58.4%) [30]. AI teens had the lowest decrease in pregnancy amongst females aged 15 to 19 in 2012, among their counterparts [31]. In addition, evidence suggests that AI youths initiate sex at earlier ages when compared to other ethnic groups, with one study on AI finding that 70% of 16 to 18 year olds and 42% aged 13 to 15 reported previous sexual intercourse [32], compared to a national study that found that 32% of 15 to 17 year olds had ever had sex [33]. Evidence also shows that sexually active AI youths are less likely than other subpopulations of youths to use contraception [34,35].

A community needs assessment that ran parallel to the CHOICES work with AIAN adult women highlighted the need to reduce the risk for AEP in AIAN adolescent girls [36]. The study noted different themes that were highlighted by the community in order to expand the CHOICES program on Native communities. The community urged to start the prevention efforts at a younger age; they emphasized the role of education in preventing AEP, and that the role of both family and culture were paramount in AEP prevention efforts. Overall, the study portrayed the community-defined importance of AEP prevention efforts. To meet this need, the CHOICES for American Indian Teens (CHAT) Program was developed to adapt the evidence-based CHOICES intervention for use with AIAN teen girls. The CHAT team aimed to gather additional community input to revise and adapt the CHOICES curriculum and to pilot test this curriculum with female AIAN adolescents aged 14–18 [36].

This paper describes the iterative process adopted by the CHAT team in designing a culturally-appropriate AEP prevention program for AIAN teens. This process utilized qualitative data stemming from key informant interviews with community members. Furthermore, the team also conducted focus groups with teens to understand what modifications should be made to the existing CHOICES curriculum and also to understand teens’ views on pregnancy prevention, lack of support from parents and guardians in discussing birth control and suggestions to include boys within the AEP prevention program. The paper presents the recommendations made by community members and teens within the community with regards to the CHOICES curriculum. In doing so, the paper discusses the community perception of AEP prevention for AIAN teens. We intend to understand what components are prioritized by this community for designing an efficacious AEP intervention for AIAN teens. It should also be noted that the data represents the suggestions and feedback of one sample of the AIAN community and not a nationally representative sample of the AIAN communities. However, we believe that the implications from our study will be useful for the future adaption of culturally-relevant and age-appropriate AEP prevention programs for teens from other communities.

## 2. Materials and Methods

This study obtained institutional review board approval from the tribe as well as from the principal investigator’s institution. All key-informant interviews (KIIs) and focus groups were conducted within the reservation by the CHAT team. Formative research began by conducting KIIs with community members, including elders, healthcare workers and counsellors who were actively involved with youth in the community. Two staff members, along with the principal investigator, travelled to the community to conduct KIIs and focus groups. In order to make it convenient for the interviewees, they were allowed to choose a time and place for the KII. The staff members travelled to the location at the time determined by the interviewees. The KIIs contained questions about FASD and asked respondents about the acceptability of CHOICES among the teens of the community. In this regard, they were asked to provide feedback on the CHOICES curriculum as well as on different activities within the program, such as maintaining a daily journal, birth control activities, along with knowledge about alcohol and drink sizes, which teens would complete as part of the program. The KIIs were conducted until the information received from the community members was saturated. Saturation occurs when the same information is being repeated by different members of the community [37]. In this case, the CHAT team collectively decided when the saturation within the key informant interviews occurred. 

After the development of the CHAT curriculum, AIAN teens from across the reservation were recruited to participate in focus groups. AIAN teens and adolescents provided feedback on the curriculum during the focus group discussions. The focus group was conducted at the tribal health office. Overall, the CHAT team conducted three focus groups. Each session lasted about an hour and was facilitated by female CHAT team members, as sensitive topics (i.e., alcohol consumption and contraceptives) were discussed. The team decided that female AIAN teens and adolescents would be comfortable in discussing topics related to birth control and alcohol consumption with a female team member. Written consent was obtained from all focus group participants, including parental consent, before each focus group. The participants were offered a $25 visa gift card for their time. The team also provided refreshments to the participants at each focus group.

Two team members with qualitative data analysis training conducted a data analysis. Coders looked for recurring themes that centered on suggestions to modify the existing CHOICES curriculum. The primary coder began the analysis by carefully analyzing the transcripts for the emergent themes. This process involved a line-by-line coding looking for new patterns. The primary coder used the repetition technique, which allowed the coder to identify a recurring pattern within the data. We assumed that the more frequently a concept occurred in a text, the more likely it was to be a theme [38]. Afterwards, the secondary coder also analyzed the data using the same emergent themes. Furthermore, the secondary coder also looked for new themes. The secondary coder also looked for themes that characterized the experience of the respondents. Examining the context, the perspectives of the respondents and their ways of thinking about people, objects, processes, events and relationship also helped to derive interesting themes from the data set [39]. In order to maintain the rigor and methodological coherence of the study, both coders discussed the emergent themes they discovered with the principal investigator (PI), and discrepancies between the coders were resolved with the help of the PI. Both coders, along with the PI, determined the final emergent themes. The qualitative data were analyzed using the textual analysis software Nvivo (version 11; QSR International Pty Ltd., Melbourne, Australia).

## 3. Results

### Demographics

A total of 15 KIIs were conducted with both women (n = 11) and men (n = 4) who ranged in age from 27 to 78 and were AI (n = 14) or African American (n = 1). All were individuals working within the reservation community or with AIAN in an off-reservation community nearby. Additionally, three focus groups were conducted with n = 15 AIAN girls (ages 15 to 19 years). Most of the girls (n = 10) were currently enrolled in high school, three girls had high school diplomas but were unemployed and two girls had no high school diploma and were also unemployed. Based on the feedback from the KIIs and focus groups, the following modifications were recommended.

*Make information appealing to teens:* This was an overarching theme that surfaced throughout the KIIs and focus groups. Suggestions included using language for teens, graphics and pictures of alcohol that were popular amongst teens in the community. The readability of the curriculum was a primary concern among the KII respondents. Most expressed concern that the curriculum would not be useful if the teens were not able to understand it. One KII respondent stated “you have to really do things at an eighth grade reading level for them to comprehend.” 

Most KII respondents also emphasized the use of graphics and pictures to convey the message of the program. Many respondents also expressed that the teens should be able to relate to the pictures used in the curriculum. They suggested using pictures of vodka and malt liquors, such as *Joose*, *Mike’s Hard Lemonade*, etc., which were reported to be popular among teens in the community. One KII respondent outlined “… including maybe like the cans that they can actually buy, that’s what they’re actually drinking. So maybe making it more realistic and leaving this but maybe saying this is equivalent to one can of… I don’t know what do they say? *Joose* or stuff like that. The things that they’re actually drinking because they don’t actually drink, you know, a cup of beer or a glass with ice.”

Another recurring suggestion was the inclusion of information about bootlegged alcohol in the curriculum. There was a growing concern about the rise of bootleg alcohol in the community, made available in clear plastic water bottles. Many interviewees, as well as focus group participants, expressed that the water bottle should be included within the curriculum because it was popular among teens in the community. One suggestion included “I think that there’s also a culture of bootlegging on Pine Ridge and so one of the missing pictures here is just even a water bottle. Because what the young people do now is they’ll get just a regular water bottle and the bootlegger will mix it half water, half vodka.”

Most of the key informant interviewees stated that the alcohol consumption pattern of teens in the reservation is different and that that should be depicted in the curriculum. As stated by one interviewee, “First of all they probably don’t drink beer from a stein. They drink it from the bottle. You know. They wouldn’t identify with that.” Additionally, they also recommended to include different types of alcohol popular among teens in the community. 

It is also important to note that the recommendations and modifications suggested during the KIIs were also echoed by the teens during the focus group discussions. Specifically, focus group participants agreed with KII respondents on the inclusion of graphics of different types of alcohol consumed by the teens in the community and the growing prominence of bootlegged alcohol in the community. 

*Information on Birth Control:* KII respondents suggested using pictures and graphics of various birth controls. One respondent stated, “There should be lot more graphic obviously. You know. And we have to go with that and not act shocked or anything like that.” Many respondents felt it was necessary to use language and graphics that are readily understood by teens. They believed that teens are not exposed to different types of birth control and that it was necessary to provide accurate information in a manner that they would understand. Since this was a sensitive topic, they also felt that teens could shut down the conversation if the information was not conveyed in an appropriate manner.

Many KII respondents believed that the lack of use of birth control in their community was not due to the accessibility to birth control but rather due to a lack of knowledge. One interviewee summarized, “I think a lot of lack of contraception is not lack of access. Because I think there’s a lot of access to it, especially in tribal communities probably more so than if you were a young teen in Rapid City.” They also felt that more could be done to spread information about the availability of birth control methods in their schools and hospitals. 

Some KII respondents suggested that teens might be uncomfortable to speak about birth control due to the stigma attached with using birth control. One respondent noted, “Some girls think, “If I go up to the hospital and I go over there, they’re going to think I have an STD (Sexually Transmitted Disease) or I’m pregnant.” So talking them through the stigma and that, you know, getting them comfortable is important.” Others also echoed that teen girls feel their parents or elders often conflate the idea of using birth control with being sexually active. Therefore, it is difficult for teen girls in the community to get proper guidance when it comes to exploring different birth control options. This issue was further validated by the teen girls during the focus group discussions. They mentioned it was difficult to discuss birth control at home and get permission to use birth control. Some of the girls mentioned that their parents and guardians were not aware of them being on birth control. The girls also emphasized the importance of prenatal care among the teens and adolescents including information on the harmful effects of alcohol on the fetus. One focus group participant explained, “I mean, I’d say, just birth control and then, I don’t know. Like, getting prenatal care, you know, if you’re pregnant. Not do those things. Like, a lot of people don’t get prenatal care and that’s when their babies end up all, like, FAS or whatever they decide to do when they’re pregnant.”

*Culture:* Many KII respondents expressed that culture should be incorporated within the CHAT curriculum. They believed that culture could be used as a tool to teach the teens about the harmful effects of alcohol. Specifically, they noted the use of American Indian culture, including “Tiospaye” (*extended family and kinship*), to communicate with the teens. One respondent explained, “So I think that Tiospaye, extending this teaching to Tiospaye and identifying the Tiospaye leaders in the communities is very important.” Respondents believed that the information regarding AEP prevention could be better conveyed through the elders and the leaders of Tiospaye because the teens respect them and would listen to them. Furthermore, each Tiospaye is a highly embedded network, and the information could easily reach numerous people at once. Therefore, inviting an elder to speak with teens could be a useful way to disseminate information on AEP prevention.

Similarly, many KII respondents supported using Lakota values to talk about alcohol use and birth control to teen girls. One respondent noted, “And talk to them about the Lakota values of years ago. They are still with us but not like as strong as they used to be. And they need to know that... the little girls need to know that their body was held to be sacred.” This theme was echoed by several KII respondents as they felt it was necessary for teens to view themselves through their own cultural lens. They believed that Lakota values offered cultural capital that the teens could use to understand where they came from, in order to have a better understanding of themselves.

Focus group participants also expressed the importance of culture in their lives. This was another recommendation that both KII respondents and the focus group participants emphasized. The focus group participants noted that they were proud of their culture and wanted to pass it on to the next generation. Furthermore, they also felt that culture played a big role in the AEP prevention program. One participant stated, “But the way things are going now and how they’re gonna-look like it’s gonna end up or something, I think we just lost our way. And especially with alcoholism, it’s-it really-we did really lose our way with that because, like, there are some kids that really wanna learn beading and really wanna learn dresses, you know, and why can’t we just bring that back?”

*Confidentiality:* This was another recurring theme that was heavily emphasized during the KIIs. “Daily Journal” is an important aspect of the original CHOICES curriculum, where women recorded their drinking and sexual activities, including the use of contraception if they were sexually active [17,18]. This activity allowed the participants to self-monitor, leading to the recognition and discussion of risky behaviors with the interventionist, as well as informing future goals. Many KII respondents were alarmed that the project will be gathering sensitive information about teen drinking and sexual activities. They were concerned about teens’ confidentiality in the program. One respondent noted, “I worry about confidentiality. Are they keeping their own journal? What if they lose it and then it’s just out there? Someone sees them drop it, then their very personal information is lying on the floor depending on if they write their name on it or not, you know?” Many KII respondents recommended that the curriculum should not be maintained in a paper format. They feared that it would be unsafe and teens could simply forget it somewhere after completing it, after which others could easily find it.

Most of the KII respondents recommended translating the program into an online format with password protection, where teens could store their sensitive information. One KII respondent explained that, “So they could have a password and they feel confident that something’s locked up. The data is locked up. And because, you know, young people that’s their world. They’re all one, you know.” Another belief among the key interviewees was that an online format would be more appealing to teens. They explained that since teens will be using their electronic devices it will be easy for them to record such activities. Furthermore, it will also be easy to send reminders to them. Additionally, it was also believed that the teens would be more interested in recording their daily activities online rather than in a diary.

*Boys should be included*: This was a latent theme that surfaced during the KIIs and focus group discussions. Most of the KII respondents stated that AEP prevention should integrate boys into the program as well. They believed that risky behaviors such as drinking and unsafe sex do not occur in isolation and that boys should also participate in AEP prevention. One KII respondent stated, “I think that boys need to understand how FAS comes about because it takes two genders to reproduce so they need to definitely understand that because their children are at risk too especially as they get older.” It was clear from the key informant interviews that the respondents felt there was a disproportionate burden of AEP prevention upon girls. They stated that it would be a disservice to girls if the prevention program did not include boys.

Similarly, teen AIAN girls in the focus group discussions also expressed that the boys should also be included in the AEP prevention programs. Specifically, this theme surfaced in the focus groups during the discussion of birth control. Several participants expressed that boys should be aware of various birth control options as well. One participant explained that the boys should put themselves in girls’ shoes and understand the consequences of being pregnant. In this regard, the focus group participants expressed that both girls and boys should bear the responsibility of AEP prevention. See Table 1 for a summary of the results from the KIIs and focus groups.

## 4. Discussion

The value in this paper is to outline necessary changes to the current AEP prevention programs that are being currently tested with only adult women, in order to make them appealing and useful to adolescents. A qualitative data analysis identified several recommendations, such as: making the information regarding alcohol and birth control appealing to teens through the use of technology, engaging teens through interactive activities, and providing education and information in a way that is relatable and age-appropriate. Specific to AIAN teens, there is a need to make public health programs culturally relevant, as has been done in the CHOICES intervention. This study outlines various components that were prioritized by community members in creating a culturally-relevant and age-appropriate AEP prevention program.

The analysis of the interview and focus group data shows that there is great potential to develop interventions for adolescents by modifying existing AEP prevention programming in specific ways. As noted in the Background, there have been few studies on teen substance use during pregnancy, with one study finding that pregnant teens report elevated levels of substance use *prior to* becoming pregnant [23,24] and another reporting that pregnant teens reported a greater substance use (18.3%) compared with same-age non-pregnant peers (13.8%) [25]. While these are national statistics, we can surmise that the AEP risk for AIAN teens is as high if not higher, given the potential number of AIAN females at risk of teen pregnancy [40], paired with a high rate of risky drinking among AIAN teens [26]. The existing and evidence-based AEP prevention programming that could be utilized with AIAN teens, found to be efficacious with adult AIAN women, includes a case management system during pregnancy [41], a web-based alcohol screening, a brief intervention, and a referral to a treatment (SBIRT) intervention [42], as well as the CHOICES intervention [21,22,43,44,45].

However, these previous efforts did not include AIAN teens, meaning that our efforts to develop the CHAT Program—or CHOICES for AIAN teens—is an important one. There is both an epidemiological need and also a community-defined need for AEP prevention in AIAN youths. Findings from past studies cite the need for AEP prevention in youths, and the tribal community responding in this project agreed with the need for AEP prevention with AIAN youths [36]. A similar qualitative study with urban AIAN adolescents and adults also found that FASD prevention should consider cultural and community values and contexts [46]. In addition, a survey with AIAN youths found that students lacked knowledge about the relationship between alcohol and FAS and that there were limited prevention programs in schools [47].

Our formative and qualitative work with AIAN teens fills that need. The formative research results highlight specific modifications to CHOICES that are needed to develop a prevention program for teens. The team has collaborated with this particular tribal community since approximately 2004 and has developed several community-led research interventions [44,48,49]. For this study, we used several tenants of community-based participatory research (CBPR) in that the community defined AEP prevention for youths as a need within the community [36], and we gathered vast community input through our formative qualitative work, constructing an intervention based on community-driven data. We also wanted to meet the needs of the youth population by utilizing technology to deliver the intervention. A description of the development of CHAT as an eHealth intervention for AIAN youths will be provided in a separate manuscript. 

The study uncovered other important and surprising aspects of AEP prevention programming for AIAN youths. For instance, a major theme was a large gap in what young AIAN women were learning about pregnancy prevention and from whom they were learning it from. While schools provided some teen pregnancy programing, and the community as a whole has addressed substance use in the past, the focus group members reported getting much of their information on birth control from friends and siblings and recommended that more formal education on pregnancy prevention must be provided. Surprisingly, it did not appear that conversations about teen substance use were routinely linked to risky sexual behavior. Both the focus group and KII interviewees seemed to compartmentalize these two behaviors, similar to previous prevention efforts with AIAN teens, which target either substance use [50,51,52,53,54,55,56,57] *or* contraception/sexual activity [58,59,60,61,62,63] separately; this is problematic as alcohol consumption can have an impact on the consistency of birth control utilization [64]. Therefore, studies on preconception health, like CHAT and others, are important in altering the community’s perception of how to prevent AEP and FASD by focusing on the dual-behavioral approach prior to pregnancy.

### Limitations

There are several limitations to this formative research study that reports on the process of developing the CHAT curriculum. The project collected data from one tribal community in the Northern Plains and was a small, qualitative study. However, studies such as these with a significant community input are important in informing the design and implementation of an intervention such as the CHAT Program. The results add to the literature on how to develop public health programs with Native communities and outline the next steps in understanding and ultimately preventing the risk of AEP in younger populations.

## 5. Conclusions

It has been well-established that FASD prevention can extend beyond pregnant women to encourage alcohol abstention. Since the early 2000s, the focus began to shift to women during the preconception period to reduce drinking in women who are at risk of an unintended pregnancy [14,16,19,20,65,66]. Likewise, preconception prevention can now move to younger, but still at-risk, populations. While much is still unknown about the epidemiology of AEP in teens, it is evident that AEP prevention programs must incorporate adolescents. The dual-behavioral approach is a modern approach to prevention with teens and can tackle the integrated relationship between substance use and risky sexual behaviors in young adults. With the addition of community input, a targeted intervention on AEP prevention with AIAN youths is now ready to be tested with at-risk teens. 

## Figures and Tables

**Table 1 ijerph-16-01795-t001:** Summary of Results.

Category	Definition	Examples
Information on Birth Control	Suggestions on how to present information on contraceptives, including readability, photos and videos, where to access credible information, and addressing stigma.	“I think a lot of lack of contraception is not lack of access. Because I think there’s a lot of access to it, especially in tribal communities probably more so than if you were a young teen in Rapid City.” “Some girls think, “If I go up to the hospital and I go over there, they’re going to think I have an STD or I’m pregnant.” So talking them through the stigma and that, you know, getting them comfortable is important.” “I mean, I’d say, just birth control and then, I don’t know. Like, getting prenatal care, you know, if you’re pregnant. Not do those things. Like, a lot of people don’t get prenatal care and that’s when their babies end up all, like, FAS or whatever they decide to do when they’re pregnant.”
Culture	Integration of American Indian culture into the program, including “Tiospaye” (*extended family and kinship*).	“And talk to them about the Lakota values of years ago. They are still with us but not like as strong as they used to be. And they need to know that... the little girls need to know that their body was held to be sacred.” “But the way things are going now and how they’re gonna-look like it’s gonna end up or something, I think we just lost our way. And especially with alcoholism, it’s-it really-we did really lose our way with that because, like, there are some kids that really wanna learn beading and really wanna learn dresses, you know, and why can’t we just bring that back?”
Confidentiality	Concerns of privacy of information obtained and suggestions on how to address them.	“I worry about confidentiality. Are they keeping their own journal? What if they lose it and then it’s just out there? Someone sees them drop it, then their very personal information is lying on the floor depending on if they write their name on it or not, you know?”
Boys should be included	Inclusion of boys into CHAT or a similar program.	“I think that boys need to understand how FAS comes about because it takes two genders to reproduce so they need to definitely understand that because their children are at risk too especially as they get older.”

STD: Sexually Transmitted Disease; FAS: Fetal Alcohol Syndrome; CHAT: CHOICES for American Indian Teens.

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
