# Peer review of "Community Perceptions of Alcohol Exposed Pregnancy Prevention Program for American Indian and Alaska Native Teens"

_ijerph, 2019, doi:10.3390/ijerph16101795_

Round 1

Reviewer 1 Report

The informative study benefits the prevention of FASDs by outlining a series of components prioritized by community members in creating a culturally-relevant and age-appropriate AEP prevention program and providing community perceptions of AEP prevention for the teens. For the results, the authors should add a table or scheme graphics to summarize the content. 

Author Response

Please see the letter attached here with for a detailed response to the reviewers.

Reviewer 2 Report

Dear Editor,

Thank you for the opportunity to review the manuscript titled "Community perceptions of Alcohol Exposed Pregnancy prevention program for American Indian and Alaska Native teens." I read the manuscript with much interest and enthusiasm. I find the study interesting and a useful piece of formative research. I herewith submit my review comments to improve the quality of the manuscript. Hope the authors will find the same helpful.

Thanking you,

Best regards,

Kavitha

Reviewer's Comments:

1.      The introduction could be reduced and could be focussed to the prevalence and enlighten the importance of such a study. In the present form, the section appears to be fluid and less focussed.

2.      The Methodology needs to be strengthened on:

1.      How transcripts were coded and analysed

2.      Theory used to explain these concepts and make sense of the transcripts

3.      When there were discrepancies between the two coders and analysers how this was resolved.

3.      A summary of the major findings at the end of results would be good to relate the same to the discussion.

4.      Discussion section could focus only on the important results from the study and the rest could be deleted. In fact this section should contain the potential interventions identified from the FGDs and a discussion on the feasibility.

5.      The manuscript could be beneficial to the readers with these inputs.

Author Response

(The authors gave the same response as above.)

Round 2

Reviewer 2 Report

May be accepted.